# Microbial Persistence, Replacement and Local Antimicrobial Therapy in Recurrent Bone and Joint Infection

**DOI:** 10.3390/antibiotics12040708

**Published:** 2023-04-05

**Authors:** Bernadette C. Young, Maria Dudareva, Margarete P. Vicentine, Andrew J. Hotchen, Jamie Ferguson, Martin McNally

**Affiliations:** 1Bone Infection Unit, Nuffield Orthopaedic Centre, Oxford University Hospitals, Oxford OX3 7LD, UK; 2Nuffield Department of Medicine, University of Oxford, Oxford OX3 9DU, UK; 3Nuffield Department of Orthopaedics, Rheumatology and Musculoskeletal Sciences, University of Oxford, Oxford OX3 7LD, UK

**Keywords:** microbiology, recurrent infection, prosthetic joint infection, osteomyelitis, fracture-related infection, local antimicrobials, antimicrobial resistance

## Abstract

We report microbiological results from a cohort of recurrent bone and joint infection to define the contributions of microbial persistence or replacement. We also investigated for any association between local antibiotic treatment and emerging antimicrobial resistance. Microbiological cultures and antibiotic treatments were reviewed for 125 individuals with recurrent infection (prosthetic joint infection, fracture-related infection, and osteomyelitis) at two UK centres between 2007 and 2021. At re-operation, 48/125 (38.4%) individuals had an organism from the same bacterial species as at their initial operation for infection. In 49/125 (39.2%), only new species were isolated in culture. In 28/125 (22.4%), re-operative cultures were negative. The most commonly persistent species were *Staphylococcus aureus* (46.3%), coagulase-negative Staphylococci (50.0%), and *Pseudomonas aeruginosa* (50.0%). Gentamicin non-susceptible organisms were common, identified at index procedure in 51/125 (40.8%) and at re-operation in 40/125 (32%). Gentamicin non-susceptibility at re-operation was not associated with previous local aminoglycoside treatment (21/71 (29.8%) vs. 19/54 (35.2%); *p* = 0.6). Emergence of new aminoglycoside resistance at recurrence was uncommon and did not differ significantly between those with and without local aminoglycoside treatment (3/71 (4.2%) vs. 4/54 (7.4%); *p* = 0.7). Culture-based diagnostics identified microbial persistence and replacement at similar rates in patients who re-presented with infection. Treatment for orthopaedic infection with local antibiotics was not associated with the emergence of specific antimicrobial resistance.

## 1. Introduction

Orthopaedic infections such as prosthetic joint infection (PJI), osteomyelitis and fracture related infection (FRI) are relatively uncommon, however treatment is difficult and clinical recurrence occurs in a substantial proportion of those treated [1,2]. A broad range of microorganisms can cause orthopaedic infection [3,4] and the microbiology of infection—including the frequency of multi-resistant organisms—changes over time [5,6].

Infection may recur even many years after the index episode, and the microbiology of recurrence is not well described. The relative contribution of relapse and re-infection is important in understanding whether treatment strategies should target microbial persistence or host susceptibility [1]. Microbes identified at index and repeat episodes can be used to understand the relative rates of relapse (with microbial persistence) or re-infection (with microbial replacement). One large meta-analysis of PJI had microbiological data available for only 41 instances of recurrent infection, and found no association between organisms cultured in PJI and chance of recurrence after revision surgery [7]. Another meta-analysis of PJI treated by debridement and implant retention (DAIR) only described microbes to the level of Gram positive, Gram negative, *S. aureus*, or MRSA [8]. Among 292 patients undergoing revision for chronic knee PJI, of whom 30 had septic failure, no organisms were significantly associated with septic failure [9]. Conversely a case series of treated extremity osteomyelitis reported that *Pseudomonas* infection was associated with higher rates of recurrence [10], and culture negative PJI was associated with better outcomes at two years in a large cohort study [11]. How microbiology from index surgery relates to subsequent surgeries has not been presented for large cohorts. One study of 92 recurrent prosthetic joint infections found re-infection was more common than relapse [12]. In a German cohort of 63 patients undergoing revision for PJI, recurrent infection was seen in 11/63, half of these with the same organism [13].

The use of local antimicrobial treatment in orthopaedic infection is attracting increasing attention, with the publication of successful case series [14,15,16] and the conduct of a randomised clinical trial of their efficacy [17]. Local antibiotics offer the attractive prospect of highly effective, targeted antimicrobials with prolonged treatment to target persisting microbes with reduced systemic exposure and toxicity [18,19,20]. However, this treatment strategy will have limited utility if persisting microbes have only a small contribution to overall recurrent infections. There is also concern that local antimicrobials may provide further selection pressure, favoring antimicrobial resistance, especially if low levels of antibiotic are eluted over a prolonged period in vivo [19].

The aim of this study was to quantify the rates of microbial persistence and replacement found at repeat surgery in a large group of patients re-presenting with confirmed orthopaedic infection. Further, we used this cohort to investigate the potential impact of local antibiotic use on antimicrobial resistance in all recurrent infections.

## 2. Results

### 2.1. Patient Cohort

137 individuals with recurrent orthopaedic infection were identified, and 125/137 (91.2%) had a minimum dataset available (Table 1 and Table 2). All patients were treated at one of two institutions with specialists in the treatment of orthopaedic infection. The median time between operations was 217 days (IQR 91–572 days, range 7–3906 days).

All patients had previously undergone treatment that was intended to be definitive for their infection. Surgical treatment included: debridement and implant retention (DAIR), single stage or two-stage revision for PJI; DAIR, revision or removal of metalwork for FRI; and surgical debridement for osteomyelitis. Local antibiotics were introduced at the time of operation according to the treating surgeons’ decision. Concomitant systemic antibiotics (intravenous with or without early oral switch) were given in all cases, guided by the results of surgically collected samples for culture.

Patient characteristics (Table 1) show a small majority of patients were male, and that PJI was the most common infection treated. Almost all patients with fracture related infection or osteomyelitis received local antimicrobial treatment.

### 2.2. Microbiological Results at Index and Recurrence Surgery

Culture results at index procedure showed the most frequently isolated species were Staphylococci (Table 2). A single species was isolated in 75/125 (60%) and multiple species in 36/125 (28.8%). A similar distribution of organisms was identified at surgery for recurrence (Table 2), with no statistically significant differences in the number of patients with each organism identified at first or second operation apart from a higher number with Enterococci identified at index surgery (19/125 vs. 8/125, *p* = 0.04, chi-squared test). A single species was isolated in 75/125 (60%) at surgery for recurrence. Fewer cases had mixed cultures at repeat operation (index operation 36/125 (28.8%) vs. repeat 22/125 (17.6%), *p* = 0.05, chi-squared test), while more cases were culture negative at repeat operation (index operation 14/125 (11.2%), vs. repeat 28/125, (22.4%), *p* = 0.03, chi-squared test).

For each instance of a species being isolated at the index procedure, the number with the same, different or no organisms isolated at the next procedure were reviewed (Table 3). In addition to the species isolated in culture, the antibiograms were reviewed to assess whether the organism was strongly suggestive of microbial persistence. Up to two differences in the reported antimicrobial susceptibilities were deemed similar (and consistent with microbial persistence). While the overall frequency of isolation of organisms was similar in index and recurrence surgeries (Table 2), there was substantial variation when these were considered at an individual level (Table 3).

At re-operation 48/125 (38.4%) individuals had an organism from the same species or group as at the index operation. The highest rates of persistence at the species level were seen in *Staphylococcus aureus* (19/41 (46.3%)), Coagulase-negative staphylococci (8/16 (50%)), and Pseudomonas species (4/8 (50%)). In 49/125 (39.2%), all organisms isolated at re-operation were different species from those grown at first operation. In 28/125 (22.4%), re-operative cultures yielded no growth.

Species identification within groups and antimicrobial susceptibility results were used to assess whether the isolation of the same species or group was highly likely to represent persistent infection (Appendix A). A total of 33/125 (26.4%) of recurrent infections met this stronger threshold for likely persistence. *S. aureus* and *Pseudomonas aeruginosa* showed the highest levels of persistence (41.5% and 37.5%), with no other organisms or groups showing strong evidence of persistence in more than 25% of cases.

### 2.3. Use of Local Antibiotics and Antimicrobial Resistance

Local antibiotics were used in 74/125 (59.2%) of patients (Table 4). Local antibiotic use was less common in treatment for PJI. Agents used were gentamicin 53/125 (42.4%), tobramycin 18/125 (14.4%), and vancomycin in 19/125 (15.2%). Combined gentamicin and vancomycin usage was seen in 16/125 patients (12.8%). No patients received dual local aminoglycoside therapy.

In this cohort of recurrent infections it was common to identify organisms that were not susceptible to Gentamicin. We define Gentamicin non-susceptibility to include both species with intrinsic Gentamicin resistance (e.g., *Bacteroides* species) and isolates reported gentamicin resistant on antimicrobial susceptibility testing. At index procedure, a Gentamicin non-susceptible organism was cultured in 51/125 patients (40.8%), and found more commonly in PJI (39/76, 51.3%) than FRI or osteomyelitis (12/49, 24.5%, *p* = 0.005, chi-squared test). At re-operation the proportion with Gentamicin non-susceptible organisms was lower: 40/125 (32.0%). There was no statistically significant difference in the rate of Gentamicin resistance at re-operation comparing patients who previously received local aminoglycosides with those who had not (21/71, 29.8% vs. 19/54, 35.2% *p* = 0.6, chi-squared test).

In 48/125 (38.4%) of patients, the same species was isolated during the index and recurrence surgery. Among patients with a persistently identified species, we identified no cases with new glycopeptide resistance and seven cases with new aminoglycoside resistance arising at the second procedure. In 2/7—*S. aureus* and *E. faecalis*—aminoglycoside resistance was the only change in antimicrobial susceptibility. In 5/7, there were at least two additional changes in observed antimicrobial susceptibility. We investigated for association between new aminoglycoside resistance at recurrence and local aminoglycoside use at first surgery. A total of 3/71 (4.2%) of cases who initially received local aminoglycoside cultured an organism with new aminoglycoside resistance at recurrence. A total of 4/54 (7.4%) of those who did not receive local or systemic aminoglycoside at index surgery cultured newly resistant organisms. Thus we find no evidence to reject the null hypothesis that local aminoglycoside did not affect subsequent aminoglycoside susceptibility (*p* = 0.7, Fisher’s exact test).

## 3. Discussion

Both re-infection and persisting or relapsing infection contribute substantial but variable roles in many recurrent infections, including Tuberculosis, urinary tract infection and *C. difficile* infection [21,22,23]. Using culture-based diagnostics, this study finds that recurrent infections were almost equally likely to be persistent or re-infections. The organisms which showed the strongest evidence of persistence were *S. aureus* and *P. aeruginosa*, consistent with a strong role for biofilm in persistent infection [24]. This demonstrates the critical importance of excellent surgical clearance and targeted, biofilm-active antimicrobial therapy informed by deep tissue culture.

In this cohort, recurrence due to bacterial species not found at index infection was seen at a similar frequency to persistent infection. This may reflect a failure to grow organisms present at the index surgery, due to incomplete sampling, pre-treatment with antibiotics, or failure to culture fastidious organisms [25,26]. However, in this cohort, we had a standardised sampling technique and culture protocol which was applied at both index and recurrence surgery. Moreover, the frequency of fast growing organisms (e.g., *S. aureus*, *Enterobacteriales*) and more fastidious ones (e.g., anaerobes and diphtheroids) was not statistically significantly different at index or repeat operation. It is likely that continuing host vulnerability contributes to re-infection. For prevention of re-infection, treatment must include appropriate management of dead-space, as well as optimal soft tissue restoration, glycaemic control, and nutrition post-operatively.

While bacterial persistence was common in this cohort, we found no evidence of local antimicrobial therapy driving new resistance or selecting out gentamicin resistant organisms at recurrence, compared to those treated without local antimicrobial therapy. This finding supports recent in vitro evidence that bacteria exposed to antibiotic-loaded bone graft substitute did not exhibit decreased susceptibility or adaptation to antimicrobials [27]. The chances of evolving resistance may be reduced by both a small bacterial population (with substantial clearance of microbial burden by surgical resection), and the very high levels of antimicrobials delivered by these materials, being consistently above the mutant prevention concentration [28]. However, in our cohort, evolving aminoglycoside resistance was seen to arise in a small number of cases, both with and without local antimicrobial use. Clinicians must remain vigilant to this possibility.

Finally, while we did not find evidence that local antibiotics favor the evolution of antimicrobial resistance, we also found that aminoglycoside non-susceptible organisms were less common in recurrence, even in those treated with local aminoglycosides at the index operation. This is consistent with a finding that gentamicin resistance did not predict recurrence in a cohort of chronic osteomyelitis treated with a gentamicin loaded ceramic carrier [16]. The high local antibiotic levels may make susceptibility predictions from in vitro testing less relevant in guiding treatment choice, as these classifications are predictions based on systemic antibiotic administration [29]. In addition, it has been shown, in vitro, that high dose local antibiotics, delivered in a modern bioabsorbable carrier, are active against resistant organisms, with only *S. aureus* with an MIC above 1024 mg/L being unaffected [30]. The use of high dose local gentamicin may have contributed to the reduction in gentamicin resistant organisms at recurrence.

Further study of local antibiotics as a means of reducing systematic antibiotic use is underway in a large randomized controlled trial [17]. They offer the promise of targeted antibiotics with reduced toxicity for patients, and reduced antimicrobial pressures in healthcare systems. The absence of evidence of large scale resistance driven by these compounds is therefore important for clinicians considering possible expansion of the use of local antibiotics in treating orthopaedic infection. Further, the finding that a substantial proportion of recurrent infections are consistent with microbial persistence supports a possible role for local antibiotics in preventing infection recurrence. This should be investigated further.

This is the largest study of the microbiology of recurrent bone and joint infections reported to date. We find that a broad range of organisms are involved in recurrent infection and that prior microbiological results are not a reliable predictor of organisms identified at recurrent infection. These findings support the practice of giving broad empiric antibiotic treatment after sampling in patients with recurrent infection, until antibiotic treatment can be targeted according to new culture results. While we must continue to identify opportunities to reduce the use of broad spectrum antimicrobial agents, treating narrowly based on previous microbiological results would fail to adequately treat many patients with recurrent bone and joint infection.

A strength of this study is that all of the surgery was performed in two centres with specialist interest in bone and joint infection. Operative sampling techniques were standardised across the sites, where multiple, independently collected samples for culture are collected prior to the administration of peri-operative prophylactic antibiotics. This same approach was followed at both index and recurrent operations. Laboratory techniques optimised to the recovery of organisms from sterile sties and prosthetic material were also standardised between procedures. This allowed a more reliable comparison of the microbiology, without possible differences due to variable culture techniques. However, as laboratory methods do change over time, it is possible that changes in methods, such as breakpoints for reporting susceptibility, may be inconsistent over the study period. This patient cohort is limited to those who underwent operative management of recurrent infection, as the gold standard microbiological sampling is carried out operatively. Patients who declined or were not fit for surgery were therefore excluded.

To improve the power of our investigation we aggregated patients undergoing surgical management of PJI, FRI, and osteomyelitis. We believe this is justified by the important similarities between these infections, including the role of biofilm in infection, the critical role for surgical debridement as well as antibiotics in management, and the range of organisms implicated [3], so that similar factors may be implicated in recurrent infection between these groups. We did observe a lower rate of local antibiotic use in PJI during this study, and confounding from differences between infection types may affect our results. An alternative strategy would be to improve power by aggregating cases across more centres, but this would introduce potential confounding by variation in clinical and laboratory practice.

In identifying likely persistent organisms, this study is limited to species identification in culture, supported by antimicrobial susceptibility testing results. This approach was adapted from the foundational work on microbiological definitions of PJI, which used the finding of the same species with indistinguishable antibiotic susceptibility testing results to define organisms in FRI, osteomyelitis, and PJI [5,31,32]. In this study we allowed for up to two changes in antimicrobial susceptibility testing results, in recognition of the fact that time and previous treatment may change susceptibility even in a clonally persisting organism. Studies of *S. aureus* outbreaks have demonstrated that an identical antibiogram was neither sensitive nor specific in identifying whether two isolates were involved in direct transmission [33]. We expect this measure to be imperfectly accurate in defining persistence. Finally, culture results may be affected by antimicrobial treatment. Our centres have a practice of stopping antibiotics for 14 days before surgery for orthopaedic infection, when it is safe to do so, but some patients will have received systemic treatment prior to surgery.

The gold standard to identify true bacterial persistence would include genotyping of organisms identified at both episodes. To carry out such a study on a large scale would require at least prospective collection from an enormous number of patients, as recurrence only affects a minority, and can occur years later. The present cohort spans 14 years, and prospective organisms isolated from orthopaedic infection were not prospectively stored in our centres, so genomic analysis was not possible in the present study. Well-constructed prospective biobanks of orthopaedic infection would provide a valuable study tool to improve our understanding of recurrent infection.

Metagenomic analysis of samples at both episodes offers the potential to both genotype organisms at first and second episodes, and also to recover organisms not viable in culture, including fastidious organisms or microbes affected by antibiotic treatment [26]. Such studies would be a powerful way to understand the relative importance of persistence and reinfection, and may reduce the impact of pre-treatment with antibiotics on diagnostic yield. Such studies have shown promise in diagnosis of orthopaedic infections [26]. If these methods are integrated into routine clinical practice they are likely to greatly enhance our understanding of not only the microbiology of infection, but the relative role of persistence and replacement in recurrent infection.

## 4. Materials and Methods

Patient population: We identified 137 patients with confirmed recurrent orthopaedic infection from two centres in the UK. Patients between 2007 and 2021 were included with prosthetic joint infection, osteomyelitis, and fracture-related infection. Cases with suspected recurrence were identified from a previous randomised clinical trial [34], prospective cohort studies [16,35] and from a further prospective cohort of patients undergoing surgery for orthopaedic infection. These studies received either NHS Health Research Authority Ethics Approval (REC 13/SC/0016, REC 20/LO/0140) or Institutional Governance review approval (OUH 2022/7657).

A retrospective review of patient notes was undertaken to confirm that a recurrence had been identified. All recruited patients had undergone the primary surgery for infection and the surgery for recurrence within our institutions. A total of 12 patients were excluded as records of the management of their recurrent infection could not be recovered, and these were excluded from further analysis.

All patients had surgical treatment that was deemed definitive for infection. In PJI this included debridement and implant retention (DAIR), single stage or two-stage revision, or excision arthroplasty. If a two-stage revision was performed, the first stage was considered index procedure. In FRI surgical management included debridement and implant retention (DAIR), revision of metalwork or removal of all metalwork. For osteomyelitis operative management was a definitive debridement. Choice of operative management was made by the treating surgeon. Usual practice in our centres is to stop antibiotics for at least 14 days prior to surgery.

All patients received medical management with antibiotics, starting with broad empiric therapy at the time of procedure (e.g., glycopeptide plus a beta-lactam with anti-pseudomonal activity), followed by pathogen directed therapy guided by culture results. No patient received empiric systemic aminoglycosides.

Case definition: As in a recent RCT [34], recurrence was determined with the finding of at least one of

Clinical: operative finding of pus at the site of bone or prosthesis OR finding of sinus tract going to prosthesis/boneMicrobiological: two or more deep tissue samples with indistinguishable microorganisms (same species, or genus if not identified to species level, with no differences in reported antimicrobial susceptibilities). Tissue specimens must be harvested in theatre with separate sterile instruments.Histological: characteristic inflammatory infiltrate or microorganisms seen

Data collection: Electronic patient records were reviewed to confirm demographic data (sex, age at time of index surgery) and details of operative procedures (dates of index and follow up procedure, nature of index procedure, and local antibiotics used at index procedure) and antimicrobial therapy (initial, empiric treatment and definitive, targeted treatment). Microbiological results from index and recurrence surgery were recorded, including every species identified in each sample, and the reported antimicrobial susceptibilities for each species.

Laboratory methods: All patients included in this cohort underwent surgery and deep tissue sampling with a minimum of five samples for microbiological investigation [31,32]. These were cultured according to a standard protocol as previously described [34]. Briefly, intraoperatively collected deep tissue samples or explanted prosthetic material were collected with separate sterile instruments and placed into individual sterile containers. Explanted material was sonicated in sterile saline [36]. Tissue specimens underwent homogenisation in sterile saline with glass beads, and this was used to inoculate liquid culture medium (aerobic and anaerobic Bactec blood culture bottles, Becton Dickinson), which were incubated at 37 °C for up to 10 days [37]. Sub-cultures were made from bottles flagging positive, inoculating whole blood agar, lysed blood “chocolate” agar, chromogenic agar (CHROMagar orientation agar), and Colombia agar with colistin and nalidixic acid (CNA agar). Subculture plates were incubated for up to 48 h in aerobic and anaerobic conditions. If available, explanted prostheses underwent sonication in sterile saline, and an aliquot inoculated to whole blood and chocolate agar for culture for 5 days aerobically, as well as to blood agar for culture up to 10 days anaerobically. All organisms isolated were identified to species level by MALDI-TOF, with the exception of coagulase-negative Staphylococci, which were reported to species level only for *S. epidermidis* and *S. lugdunensis*. Others were reported as coagulase-negative Staphylococci.

Antimicrobial susceptibility testing was performed according to local standard operating procedure (by disc diffusion, MIC or automated methods) and recorded as susceptible, intermediate, or resistant (S/I/R) according to clinical breakpoints in use at the time of reporting. These were initially British Society Antimicrobial Chemotherapy breakpoints, subsequently replaced by EUCAST breakpoints following harmonization between these methods in 2016.

Tissue for histopathology was embedded in paraffin and cut into 5 µm sections. Sections were stained with haematoxylin and eosin, as well as Gram stained, and at least 10 high power fields (×400) were examined in each section. The finding of five or more neutrophils per high power field was diagnostic of infection [38,39].

Antimicrobial resistance classification: In addition to laboratory reported resistance, intrinsic resistance was inferred for the following agents and species: Enterococci were assumed to have intrinsic low level resistance to aminoglycosides (Gentamycin, Tobramycin and Amikacin); Enterobacteriales and Pseudomonas species were assumed to have intrinsic resistance to glycopeptides (both Vancomycin and Teicoplanin).

Distinguishing persistent and replacement organisms: If an individual had the same species identified at both procedures, and no more than two antimicrobials with different reported susceptibilities (S/I/R), these were deemed ‘similar’ and regarded as most likely to represent persistent organisms. For coagulase-negative Staphylococci genus level was accepted, as species level was not otherwise reported. If a different species was cultured at recurrence (i.e., not present at index surgery), this was regarded as a replacement organism, and most likely to represent new or re-infection.

Statistical analysis: Statistical analyses were conducted in R v4.1.0. Differences in proportions between groups for categorical data were performed using chi-squared tests (or Fisher’s exact test for comparisons including any value below five).

## 5. Conclusions

Re-infection with different organisms was seen at similar rates to persistent infection with the same species in this cohort. *Staphylococcus aureus* and *Pseudomonas aeruginosa* are the organisms most likely to be persistently identified in recurrent infections. As a group, patients whose treatment for orthopaedic infection included local antibiotics did not exhibit higher rates of specific antimicrobial resistance compared with those not treated with local antibiotics. However, we did identify a few cases where bacteria developed aminoglycoside resistance regardless of their initial antimicrobial therapy. This should be considered in antimicrobial choice during surgery for recurrence.

## Figures and Tables

**Table 1 antibiotics-12-00708-t001:** Characteristics of patients with recurrent infection.

	With Local Antimicrobial Treatment *n* = 74	Without Local Antimicrobial Treatment *n* = 51	All *n* = 125
Age, years at surgery (Median, IQR)	59.8 (46.9–70.5)	69.8 (63.2–75.9)	64 (61.2–72.6)
Male (*n* (%)) Female (*n* (%))	46 (62.2)	32 (62.7)	78 (62.4)
	28 (37.8)	19 (37.2)	47 (37.6)
PJI (*n* (%)) FRI (*n* (%)) OM (*n* (%))	29 (39.2)	47 (92.2)	76 (60.8)
	28 (37.8)	2 (3.9)	30 (24.0)
	17 (23.0)	2 (3.9)	19 (15.2)

IQR, interquartile range. PJI, prosthetic joint infection. FRI, fracture related infection. OM, osteomyelitis.

**Table 2 antibiotics-12-00708-t002:** Frequency of organisms found at index procedure for infection and next operation for infection.

Organism/Group	First Operation	Second Operation
*Staphylococcus aureus*	41	30
*Staphylococcus epidermidis*	19	18
*Staphylococcus lugdunensis*	4	2
Other CoNS ^1^	16	14
Enterobacterales ^2^	24	21
Enterococci	19	8
Streptococci	11	9
*Pseudomonas* sp.	8	8
Diphtheroids ^3^	6	4
Anaerobic sp.	4	5
*Candida* sp.	1	3
No growth	14	28

^1^ Other CoNS = Coagulase-negative staphylococci, not otherwise speciated; ^2^ Enterobacterales includes all genera of this family, including *E. coli*, *Klebsiella* species, *Enterobacter* species, *Proteus* species, *Serratia species*, *Morganella* species; ^3^ Diphtheroid includes *Corynebacterium* species, *Cutibacterium* species. First operation is the earliest operation for infection in the cohort, and may not be the first operation the patient has ever had to treat infection.

**Table 3 antibiotics-12-00708-t003:** Similarity of organisms found at index procedure for infection and next operation for infection.

Organism/Group Found in Culture at First Operation	*n*	Same Organism/Group at Recurrence *n* (%)	Same Species and Similar Antibiogram ^4^ *n* (%)	Different Species at Recurrence *n* (%)	Culture Negative at Recurrence *n* (%)
*Staphylococcus aureus*	41	19 (46.3)	17 (41.5)	14 (34.1)	8 (19.5)
*Staphylococcus* *epidermidis*	19	8 (42.1)	4 (21.1)	4 (21.1)	7 (36.8)
*Staphylococcus* *lugdunensis*	4	0	0	2 (50.0)	2 (50.0)
Other CoNS ^1^	16	8 (50.0)	4 (25.0)	5 (31.3)	3 (18.8)
Enterobacterales ^2^	24	10 (41.7)	3 (12.5)	10 (41.7)	4 (16.7)
Enterococci	19	3 (15.8)	1 (5.3)	13 (68.4)	3 (15.8)
Streptococci	11	1 (9.1)	1 (9.1)	9 (81.8)	1 (9.1)
*Pseudomonas* sp.	8	4 (50.0)	3 (37.5)	4 (50.0)	0
Diphtheroids ^3^	6	0	0	6 (100)	0
Anaerobic sp.	4	1 (25.0)	0	1 (25.0)	2 (50.0)
*Candida* sp.	1	0	0	1 (100)	0
No growth	14	n/a	n/a	11 (78.6)	3 (21.4)

^1^ Other CoNS = Coagulase-negative staphylococci, not otherwise speciated; ^2^ Enterobacterales includes all genera of this family, including *E. coli*, *Klebsiella* species, *Enterobacter* species, *Proteus* species, *Serratia species*, *Morganella* species; ^3^ Diphtheroid includes *Corynebacterium* species, *Cutibacterium* species. ^4^ Conserved antibiogram: up to 2 different results in antibiotic susceptibility testing. First operation is the earliest operation for infection in the cohort, and may not be the first operation the patient has ever had to treat infection.

**Table 4 antibiotics-12-00708-t004:** Use of local antimicrobial agents.

Antimicrobial Agents	*n* (%)	PJI *n* = 76	FRI *n* = 30	OM *n* = 19
Gentamicin	37 (29.6)	15	15	7
Tobramycin	18 (14.4)	0	9	9
Vancomycin	3 (2.4)	3	0	0
Gentamicin plus Vancomycin	16 (12.8)	11	4	1
None	51 (40.1)	47	2	2

## Data Availability

Demographic data not made publicly available due to privacy concerns, available on request from the corresponding author.

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
