# Peer review of "Microbial Persistence, Replacement and Local Antimicrobial Therapy in Recurrent Bone and Joint Infection"

_antibiotics, 2023, doi:10.3390/antibiotics12040708_

Round 1

Reviewer 1 Report

In this manuscript, the authors reported microbiological results from a cohort of recurrent bone and joint infection to define the contributions of microbial persistence or replacement. The results of this study have reference value for orthopedic clinical practice. However, there're still several issues which should be addressed.

Does the patient have diabetes, osteoporosis and other diseases that affect bone healing? Is there a history of radiotherapy or bisphosphonate drug use?

What are the methods of bacterial culture?  Does it include anaerobic bacteria?

Was the patient treated with antibiotics before surgery? Does the use of antibiotics affect the results of bacterial culture? How to eliminate these effects.

Why don't bacteria be tested by DNAsequencing? This should be explained and discussed in the section of discussion.

The application value of local antibiotics should be further discussed and difined.

The application of the research result should be discussed in details in the section of discussion.

Author Response

Reviewer 1

Does the patient have diabetes, osteoporosis and other diseases that affect bone healing? Is there a history of radiotherapy or bisphosphonate drug use?

  • We agree with the reviewer that it would be informative to add data on patient specific risk factors that increase the chance of recurrence, including systemic disease affecting bone healing. The cohorts from which we have drawn data for this study did not systematically gather this information, so unfortunately this is not available for this report.

What are the methods of bacterial culture?  Does it include anaerobic bacteria

  • We have expanded this area of methods to describe in more detail, including anaerobic culture. (Methods, line 398-407)

Was the patient treated with antibiotics before surgery? Does the use of antibiotics affect the results of bacterial culture? How to eliminate these effects.

  • Data on prior antibiotic treatment was not systematically gathered in the studies from which we have drawn this cohort, so this information is not systematically available. However we have expanded our methods section to explain the usual process in our centres of stopping antibiotics 14 days prior to surgery (line 367), and expanded our discussion to include a discussion of how future studies may reduce the impact of antibiotics on microbial detection (line 335-338)

Why don't bacteria be tested by DNAsequencing? This should be explained and discussed in the section of discussion.

  • The organisms were not stored in the long term in our centres. We have made this more explicit in our discussion (line 327-331)

The application value of local antibiotics should be further discussed and defined.

  • We have expanded our discussion to include more detail on the value of local antibiotics in treating orthopaedic infection, to give further context to our results (line 264-272)

The application of the research result should be discussed in details in the section of discussion.

  • We have expanded our discussion to put these results in the context of clinical practice (line 274-281)

Reviewer 2 Report

Dear Authors

I have reviewed your paper "Microbial persistence, replacement and local antimicrobial therapy in recurrent bone and joint infection" carefully and found it very interesting. However, I have a few following queries:

1. Why were any predisposing factors, such as diabetes, steroid therapy, immunosuppression, and foreign body implantation, not included in the patient cohort data? 

2. In my point of view, authors should also show the rate of amputation and mortality after a reoccurrence of infection, if any. 

3. Disclose the methods precisely"Antimicrobial susceptibility testing was performed according to local standard operating procedure (by disc diffusion, MIC or automated methods) and recorded as sensitive, intermediate or resistant (S/I/R) according to clinical breakpoints in use at the time of reporting"

4. Institutional Review Board disclosures should be added in the methods section. 

Author Response

  1. Why were any predisposing factors, such as diabetes, steroid therapy, immunosuppression, and foreign body implantation, not included in the patient cohort data? 

It would be informative to add data on patient specific risk factors that increase the chance of recurrence. The cohorts from which we have drawn data for this study did not systematically gather this data, sot it was not available for this report.

  1. In my point of view, authors should also show the rate of amputation and mortality after a reoccurrence of infection, if any. 

This would be helpful outcome data, especially as It would inform our understanding of the impact of recurrent infection. Unfortunately the cohorts from which we drew these patients did not all follow up after discharge to confirm these outcomes. As both centres take referrals from wide geographic areas of the UK, this information is not available in our data.

  1. Disclose the methods precisely"Antimicrobial susceptibility testing was performed according to local standard operating procedure (by disc diffusion, MIC or automated methods) and recorded as sensitive, intermediate or resistant (S/I/R) according to clinical breakpoints in use at the time of reporting"

- We have expanded this section of the methods, to give details of the of clinical guidelines used in our setting over the 14 years of this study (line 411-413)

  1. Institutional Review Board disclosures should be added in the methods section. 

- These have been added (line 347-9)

Reviewer 3 Report

This manuscript may be appreciated because of large scale  study on recurrent bone and joint infection. This reviewer suggests following points to revise the manuscript.

1. Authors used "gentamicin non-sensitive" in the text. However, this meaning is not clear. Add definition for it.

2. Materials and Methods section: Description of susceptibility test is not sufficient. Which standard did authors use to determine susceptibility? 

3. "sensitivity" should be changed as "susceptibility" throughout the manuscript.

4. line 81-83: This is repeat of the information in Table 1. Do not repeat data in Tables/Figures in main text. 

5. Table 2: How long is the period between first and second operation? If it is not written, please add this information.

6. This manuscript shows no Table of susceptibility test (even supplementary Table is not available). Readers may be interested in which bacterial species developed antimicrobial resistance in the 2nd operation. Any findings of AMR should be shown in text and/or Table/Figure.

7. Correlation of the antimicrobials used (table 4) to occurrence of drug resistance is not clear. It should be described.

8. Although this manuscript introduced interesting study, general descriptions seem to be ambiguous. Authors should revise it to focus on some specific points, to show the findings clearer.     

Author Response

  1. Authors used "gentamicin non-sensitive" in the text. However, this meaning is not clear. Add definition for it.

- We have expanded the definition included in results (lines 158-161)

  1. Materials and Methods section: Description of susceptibility test is not sufficient. Which standard did authors use to determine susceptibility? 

- We have expanded this section of the methods, to give details of the of clinical guidelines used in our setting over the 14 years of this study (line 411-3)

  1. "sensitivity" should be changed as "susceptibility" throughout the manuscript.

- This has been edited throughout (in reference to antimicrobial sensitivity, but not to test sensitivity/specificity).

  1. line 81-83: This is repeat of the information in Table 1. Do not repeat data in Tables/Figures in main text. 

- This has been edited to acknowledge between group differences rather than restate results data (lines 85-87)

  1. Table 2: How long is the period between first and second operation? If it is not written, please add this information.

- Time between surgeries added to results (line 76-77)

  1. This manuscript shows no Table of susceptibility test (even supplementary Table is not available). Readers may be interested in which bacterial species developed antimicrobial resistance in the 2nd operation. Any findings of AMR should be shown in text and/or Table/Figure.

- We have provided supplementary tables with species isolated and antimicrobial susceptibility testing results first operation (Table S1) and re-operation (Table S2)

  1. Correlation of the antimicrobials used (table 4) to occurrence of drug resistance is not clear. It should be described.

- Two paragraphs of the discussion describe the relationship between local antimicrobial use and drug resistance. We first examine whether the likelihood of isolating an organism which is not susceptible to Gentamicin at recurrence is higher in the group receiving local aminoglycoside treatment at first operation compared with those who did not. We next describes the individual cases where an organism which is not susceptible to Gentamicin at recurrence was not found at index procedure but was found at recurrence (with the same species). These are cases where aminoglycoside resistance developed in vivo. We then test the null hypothesis that those receiving local aminoglycosides were no more likely to demonstrate such resistance, and find no evidence to reject the null hypothesis.

We have revised these paragraphs to make this discussion clearer (lines 158-180).

  1. Although this manuscript introduced interesting study, general descriptions seem to be ambiguous. Authors should revise it to focus on some specific points, to show the findings clearer.   

- We have expanded our decision to more clearly elucidate how these specific findings affect clinical practice in the treatment of orthopaedic infection, both in use of local antibiotics and choice of empiric treatment (lines 264-281)

Round 2

Reviewer 3 Report

The revised version seems to be appropriately modified.